# Association between antidepressant use during pregnancy and miscarriage: a systematic review and meta-analysis

Sophie Smith [ORCID],[1] Flo Martin,[2] Dheeraj Rai,[1,3,4] Harriet Forbes[5]

SS and FM are joint first authors.

[1]Centre for Academic Mental Health, Population Health Sciences, University of Bristol, Bristol, UK
[2]MRC Integrative Epidemiology Unit, Population Health Sciences, University of Bristol, Bristol, UK
[3]NIHR Biomedical Research Centre, University of Bristol, Bristol, UK
[4]Bristol Autism Spectrum Service, Avon and Wiltshire Partnership NHS Mental Health Trust, Bristol, UK
[5]London School of Hygiene and Tropical Medicine, London, UK

**Correspondence to**
Sophie Smith;
sophie.smith@bristol.ac.uk

## ABSTRACT

**Background** Literature surrounding the association between antidepressant use during pregnancy and miscarriage is conflicting. We aimed to conduct a systematic review and meta-analysis of studies among pregnant women regarding the association between exposure to antidepressants during pregnancy and the risk of miscarriage, compared with pregnant women not exposed to antidepressants.

**Design** We conducted a systematic review and meta-analysis of non-randomised studies.

**Data sources** We searched Medline, Embase and PsychINFO up to 6 August 2023.

**Eligibility criteria and outcomes** Case-control, cohort and cross-sectional study designs were selected if they compared individuals exposed to any antidepressant class during pregnancy to comparator groups of either no antidepressant use or an alternate antidepressant.

**Data extraction and synthesis** Effect estimates were extracted from selected studies and pooled using a random-effects meta-analysis. Risk of bias (RoB) was assessed using the Risk of Bias in Non-Randomised Studies of Interventions (ROBINS-I) tool, and heterogeneity assessed using the $I^2$ statistic. Subgroup analyses were used to explore antidepressant classes and the impact of confounding by indication.

**Results** 1800 records were identified from the search, of which 29 were included in the systematic review and meta-analysis. The total sample included 5 671 135 individuals. Antidepressant users initially appeared to have a higher risk of miscarriage compared with unexposed individuals from the general population (summary effect estimate: 1.24, 95% CI 1.18 to 1.31, $I^2$=69.2%; number of studies (n)=29). However, the summary estimate decreased when comparing against unexposed individuals with maternal depression (1.16, 1.04 to 1.31; $I^2$=58.6%; n=6), suggesting confounding by indication may be driving the association. 22 studies suffered from serious RoB, and only two of the 29 studies were deemed at moderate RoB.

**Conclusions** After accounting for maternal depression, there is little evidence of any association between antidepressant use during pregnancy and miscarriage. Instead, the results indicate the biasing impact of confounding by indication.

## STRENGTHS AND LIMITATIONS OF THE STUDY

⇒ A large amount of studies were identified through an extremely thorough search strategy, thus enhancing statistical power and insight over previous reviews.
⇒ Confounding by indication and depression severity was considered in the analysis and results, elucidating the true independent effect of antidepressant use during pregnancy.
⇒ The ROBINS-I tool was used to evaluate the strength and quality of evidence.
⇒ The studies included could suffer from publication bias, and small studies producing inverse relationships are absent from the field.
⇒ High heterogeneity existed between papers, likely due to the included studies investigating different research questions, implementing different methodologies and populations, timing of exposure and confounding adjustment, thereby reducing the quality of the meta-analysis.

## INTRODUCTION

Depression affects around 264 million people worldwide and is one of the top 3 leading causes of years lived with disability globally.[1] This high prevalence is also seen among pregnant women, with 15%–23% suffering from a depressive disorder.[2] Around 3%–8% of women are prescribed antidepressants during their pregnancy in Europe.[3] This figure appears to be on an upwards trajectory since the 1990s,[4] potentially driven by both increasing prenatal depression rates,[5] and a shortage of non-pharmacological treatment options.[6]

Miscarriage, also referred to as spontaneous abortion, is defined as pregnancy loss before viability; however, the gestational threshold of viability varies between countries, from 20 to 28 weeks.[7] In the UK, viability is determined as up to 24 weeks.[7] Depending on the definition, miscarriage risk varies; on average, it occurs in approximately 15% of pregnancies.[7] Several studies and reviews found an increased miscarriage risk in women using antidepressants during pregnancy, compared with unexposed individuals.[8–10] Antidepressants can cross the placental barrier through passive diffusion,[11] highlighting biological

plausibility. Selective serotonin reuptake inhibitors (SSRIs) and serotonin-norepinephrine reuptake inhibitors (SNRIs) are the most frequently used pharmacotherapy option for depression,[12] both of which increase serotonin levels in the synaptic cleft.[3 13] Serotonergic systems are involved in embryonic developmental processes including neural crest cell differentiation,[13] a process producing cells and tissue types.[14] This highlights the importance of serotonin in embryogenesis thus its potential influence on miscarriage. Alternatively, untreated maternal depression has been shown to be associated with miscarriage.[15] This is an issue in observational studies known as confounding by indication, whereby maternal depression can independently affect the outcome of miscarriage. This could be through a number of mechanisms and mediators including lifestyle factors,[16] or stress and inflammation.[15] Including pregnant women in randomised controlled trials (RCTs) is unethical; thus, all evidence is observational and therefore susceptible to bias, making causal inference challenging. This is highlighted by studies whose observed associations between antidepressant use during pregnancy and miscarriage attenuated after stratifying for maternal depression.[17 18]

The methodological limitations that exist in previous reviews means it remains unclear whether a causal relationship exists. Therefore, clear evidence and guidance around the association between antidepressant use during pregnancy and miscarriage is important as it will allow depressed pregnant women to make well-informed decisions and reduce their decisional conflict.[19] Poor perinatal mental health is becoming an ever-growing issue, up-to-date studies are needed now which investigate child outcomes in depressed women to help drive screening, intervention and prevention strategies.

Existing reviews examining this relationship investigated several perinatal outcomes and produced differing conclusions.[20-22] These reviews have some methodological limitations including incomplete search strategies,[20] do not include more recent studies from the last 10 years,[21 22] and did not fully address confounding by indication,[20-22] a crucial element in isolating the true effect of antidepressants. This issue was addressed in the current review through using a depressed comparator, to allow for increased comparability between exposed and unexposed individuals.

The main objective of this study is to conduct a systematic review and meta-analysis of studies among pregnant women regarding the association between exposure to antidepressants during pregnancy and the risk of miscarriage, compared with pregnant women not exposed to antidepressants.

## METHODS

### Protocol registration

This study is conducted in accordance with the Preferred Reporting Items for Systematic Review and Meta Analysis (PRISMA) 2020, as well as the Cochrane handbook.[23]

The protocol for this meta-analysis was registered in PROSPERO CRD42021262839, available from https://www.crd.york.ac.uk/prospero/display_record.php?ID=CRD42021262839.

### Search strategy and information sources

We searched the databases, Embase (Ovid, 1980 to 2023), Medline (Ovid, 1946 to 2023) and PsycInfo (Ovid, 1806 to 2023) from database inception to 8 March 2021 using a comprehensive search strategy (online supplemental table S1). In brief, the following search terms were used: (antidepress* OR [generic/branded antidepressant name) AND (spontaneous abortion OR miscarriage). The search was updated on 4 August 2021 prior to commencement of data analysis, a final search before finishing the review was conducted on 21 January 2022, as well as a search to update findings before publication on 6 August 2023. Backward citation searching was conducted from papers and existing reviews; additionally, conference abstracts found in the searches were included in the review. The outcome of stillbirth was also included in our search strategy as these two outcomes are frequently reported together. However, stillbirth was not included in this review due to a higher number of relevant studies found than expected, ensuring literature was investigated at a thorough level. We placed no restrictions on language or publication status. Abstracts in a language other than English were translated using a translation software, and full texts were assessed using a translator.

### Eligibility criteria

Studies were included if the study population included pregnant women, the exposure was antidepressants, and the outcomes included miscarriage. Any antidepressant class was eligible, and no restriction was placed on timing, dose or duration of use. The comparator groups included pregnant women prescribed a different antidepressant to the primary specified type, or women with no antidepressant exposure. Studies investigating polypharmacy involving medication other than antidepressants were excluded; however, studies investigating more than one antidepressant class were included. Observational studies included used a cohort, case-control, or cross-sectional design. RCTs were included if they fit the eligibility criteria. Reviews, meta-analyses, ecological studies, case reports and case series were excluded.

### Study selection

The papers identified from the search strategy had their title and abstract screened independently by two reviewers in parallel (FM and HF). We used Endnote to deduplicate references. Outputs of the review were then cross checked, and discrepancies discussed with a third reviewer (SS). The full text of papers that passed title and abstract screening were obtained through Endnote, and three researchers completed a full-text review of a proportion of papers (SS, FM and HF). Final decisions

and discrepancies were discussed within the whole team, including a consultant psychiatrist (DR).

### Data extraction

A standardised template was used for data extraction. A patient, exposure, comparator, outcomes and study characteristic (PECOS) framework was used whereby information was collected on all variables. Variables of study design, country of study, study period, data sampling methods, type and timing of antidepressant exposure, definition of exposure and comparator groups, identification of exposure and outcome, sample size, confounding adjustment, raw data and adjusted estimates (OR, risk ratio (RR), HR) including CIs were extracted. If more than one comparator group was used, the raw data and adjusted estimates for each comparison were extracted. Data on any secondary outcomes were not extracted. Data extraction was conducted independently by two reviewers for all papers, outputs were cross checked (SS and FM), and discrepancies were discussed with a third reviewer (HF).

### Risk of bias assessment

Studies were assessed using the ROBINS-I tool.[24] This tool evaluates non-randomised studies estimating intervention effects. Seven domains were assessed, including issues arising prior to and after intervention. The ROBINS-I tool is designed for cohort studies; therefore, a template alteration specific to case-control studies was sought from the author of the ROBINS-I tool to allow for continuity in risk of bias (RoB) assessment across all study designs. RoB was independently assessed by two reviewers in parallel for all studies (SS and FM). The scoring and justification were compared, with any discrepancies discussed with a third reviewer (HF).

### Statistical analysis

Meta-analyses were performed using STATA (V.16.0). Results were presented by comparator group used, specifically general population unexposed, depressed unexposed and depressed alternative antidepressant exposure. Additionally, subgroup analysis on SSRIs and SNRIs were conducted. Adjusted effect estimates were the preferential estimate when pooling results. HR, RR and OR were treated as equivalent and pooled together, a sensitivity analysis on this assumption was conducted (online supplemental figure S1). If studies gave estimates for multiple exposure groups including differing trimester estimates, these were pooled into one result using a meta-analysis. When adjusted estimates were not provided, ORs and SEs were produced from raw data. Analyses were conducted using a random-effect model, due to the high likelihood of heterogeneity expected. Heterogeneity between studies was assessed using the $I^2$ statistic.[25] Additionally, a funnel plot and Egger test were conducted to assess evidence for publication bias.[26] A sensitivity analysis was conducted whereby papers with an overall critical RoB, or no information as identified through ROBINS-I tool were excluded. Additionally, another sensitivity analysis was done whereby only papers which used prescription databases to classify exposure were included.

### Patient and public involvement

No patient or public involvement.

## RESULTS

### Search results

The initial and final search retrieved 2010 citations. After duplicates were removed, 1800 records had title and abstract screening, resulting in a further 1751 records being excluded. The full text of 49 articles were assessed for eligibility. Articles were excluded due to reasons including incorrect study design, insufficient data or if only stillbirth was investigated (online supplemental table S2). This resulted in 25 studies being excluded, and 24 included, with the final searches adding a further 5 papers to the SR and meta-analysis[9 10 17 18 27–51]; highlighted in the PRISMA flow diagram (figure 1).[52]

### Study characteristics

Online supplemental table S3 details characteristics of the included studies. Only observational studies were found; 25 cohort, and 4 case-control studies,[10 40 50 51] and all studies were published between 1996 and 2022. The total number of subjects investigated across case-control was 314 682, compared with 5 356 453 across cohort studies. The study size varied from 247 to 1 281 418. The miscarriage definition was rarely stated, however, ranged from loss of fetus at either 20 (n=3),[10 42 45] 22 (n=2)[17 18] or 24 weeks' (n=1)[44] gestation. A mixture of comparator groups were used across studies. All studies had a comparator group of unexposed individuals without depression; some papers used additional comparator groups including depressed pregnant women exposed to an alternative antidepressant to the primary specified type,[29 37 38 43 44 48] unexposed depressed pregnant women,[17 18 37 42 43 50] and pregnant individuals using antidepressants whereby the indication for treatment was not depression.[17 32 43] Exposure status was predominantly identified through two methods. First, 16 studies used self-report to teratogen information services through either the individual or their physician.[9 27–34 36 38 40 41 43–45] Second, 14 studies used prescription databases.[10 17 18 37 39 42 47 48] Individual antidepressant types were investigated, including 12 studies examining SSRIs[18 27–29 31 34 37–39 43 45 50] and 5 SNRIs.[29 41 44 46 48]

### RoB assessment

The ROBINS-I tool found much of the literature to be at serious RoB. Online supplemental table S4 highlights that only 2 out of 29 papers had an overall moderate RoB,[44 48] compared with 22 deemed serious, and the remaining deemed critical. This tool highlighted a key area of bias being confounding. The three crucial confounding domains identified for this field were indication for

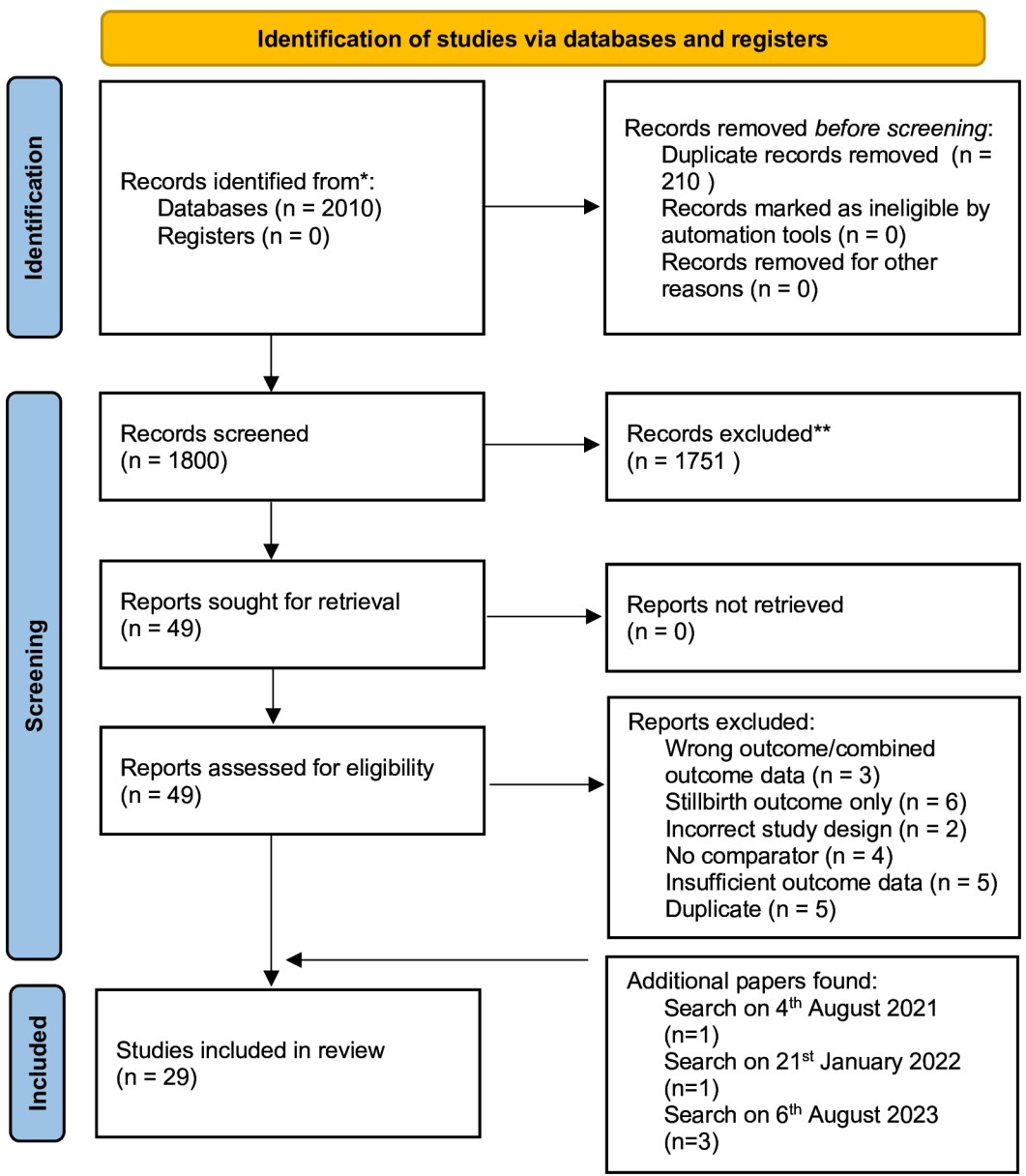

**Figure 1** Preferred Reporting Items for Systematic Reviews and Meta-Analyses flow diagram. Flow diagram illustrating the number of included and excluded studies at each stage of screening, and reasons for exclusion. This version has been adapted to include the element of additional papers found in following searches.

treatment, individual characteristics and lifestyle exposures. The confounding variables believed to be most important included presence and severity of depression, as well as gestational age. Although a large proportion of studies stratified by depression (62%), only 14% and 31% of studies accounted for depression severity and gestational age, respectively (online supplemental table S5). Confounding adjustments varied dramatically between studies suggesting a potential heterogeneity source. As very few confounding variables were included in these studies, it limits the reliability of the results of this systematic review.

### Publication bias
Publication bias was assessed through visual inspection of the funnel plot (online supplemental figure S2) and the Egger test. Asymmetry of the funnel plot indicated potential for publication bias, and the Egger test produced a p value of 0.001.

### Sensitivity analysis
Papers with an overall critical RoB or inadequate information to determine their bias category were removed (n=5). The papers removed were deemed critical as they had no confounding adjustments. Results remained very similar, with a change in the pooled estimate of 0 and 0.01 in all antidepressants and SSRI analysis, respectively, in the general population after removing the critical papers. Additionally, in the SNRI general population analysis, there was a change in the pooled estimate of 0.25 after removing the papers with no adjustments (online supplemental table S6).

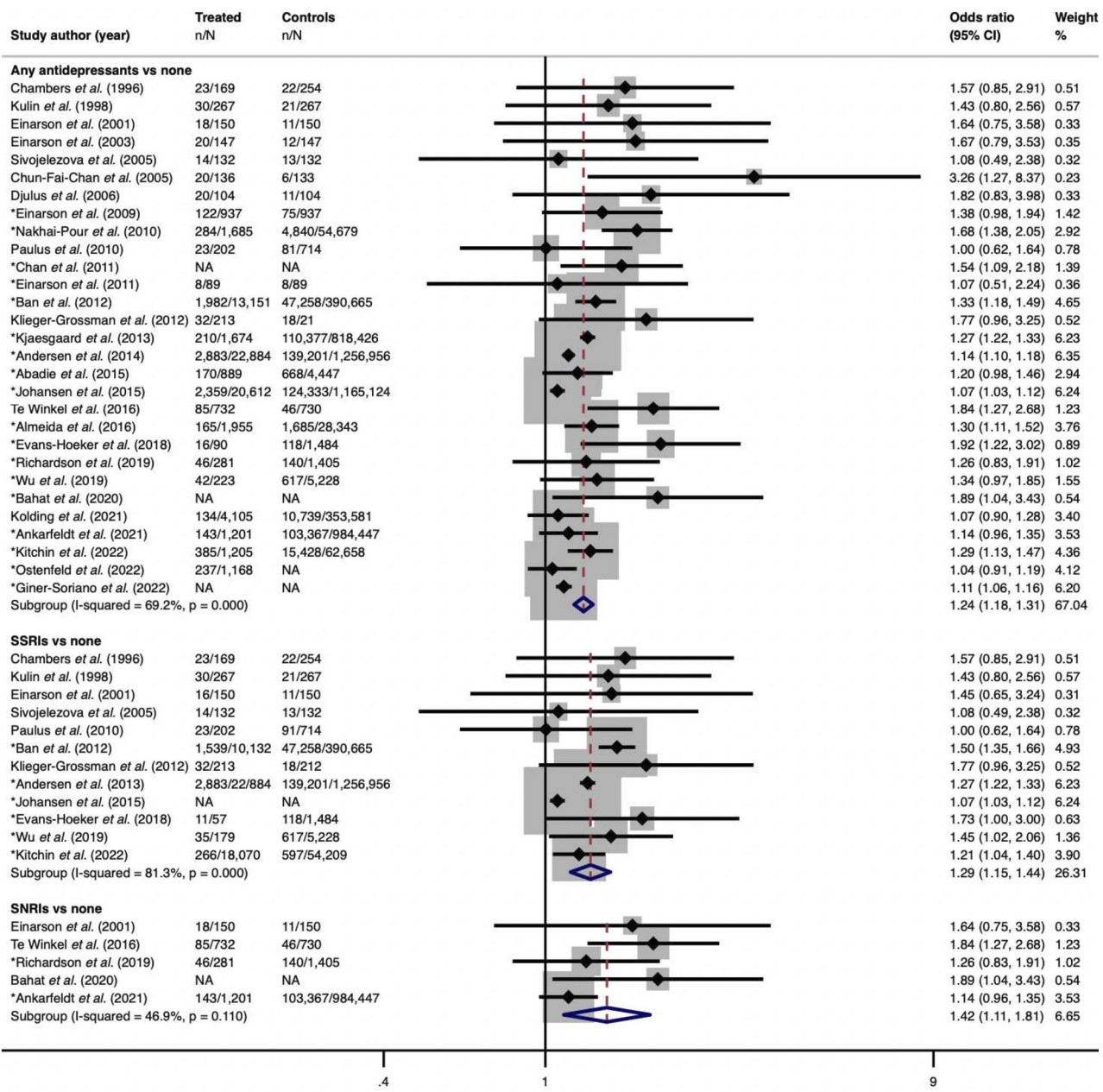

**Figure 2** Subgroup meta-analyses using general population comparator. Three subgroup analyses conduced. First subgroup assessing any antidepressant use compared with unexposed individuals in general population. Second subgroup comparing selective serotonin reuptake inhibitors (SSRIs) use and unexposed general population individuals. Third subgroup comparing between serotonin-norepinephrine reuptake inhibitor use and unexposed general population individuals.

We performed a meta-analysis using papers whereby exposure data were collected only through prescription databases. These results highlighted an attenuation towards the null compared with the primary meta-analysis using all exposure classification methods (online supplemental table S6).

### Meta-analysis results

Across all 29 studies using an unexposed general population comparator, antidepressant use was associated with miscarriage. Figure 2 illustrates the summary effect estimate (pooled OR=1.24 95% CI 1.18 to 1.31, $I^2$=69.2%,

p<0.001), which included 18 adjusted estimates. However, when stratifying for maternal depression and using a comparator of unexposed depressed pregnant women, the summary estimate decreased (pooled OR=1.16 95% CI 1.04 to 1.31, $I^2$=58.6%, p=0.034) (figure 3).

Additionally, studies were separated which investigated SSRIs or SNRIs and their possible varying risks. When comparing SSRI use and no antidepressant use, the pooled summary estimate for the outcome of miscarriage was 1.29 (95% CI 1.15 to 1.44, $I^2$=81.3%, p<0.001). Alternatively, when comparing SNRI use with no antidepressant use, a

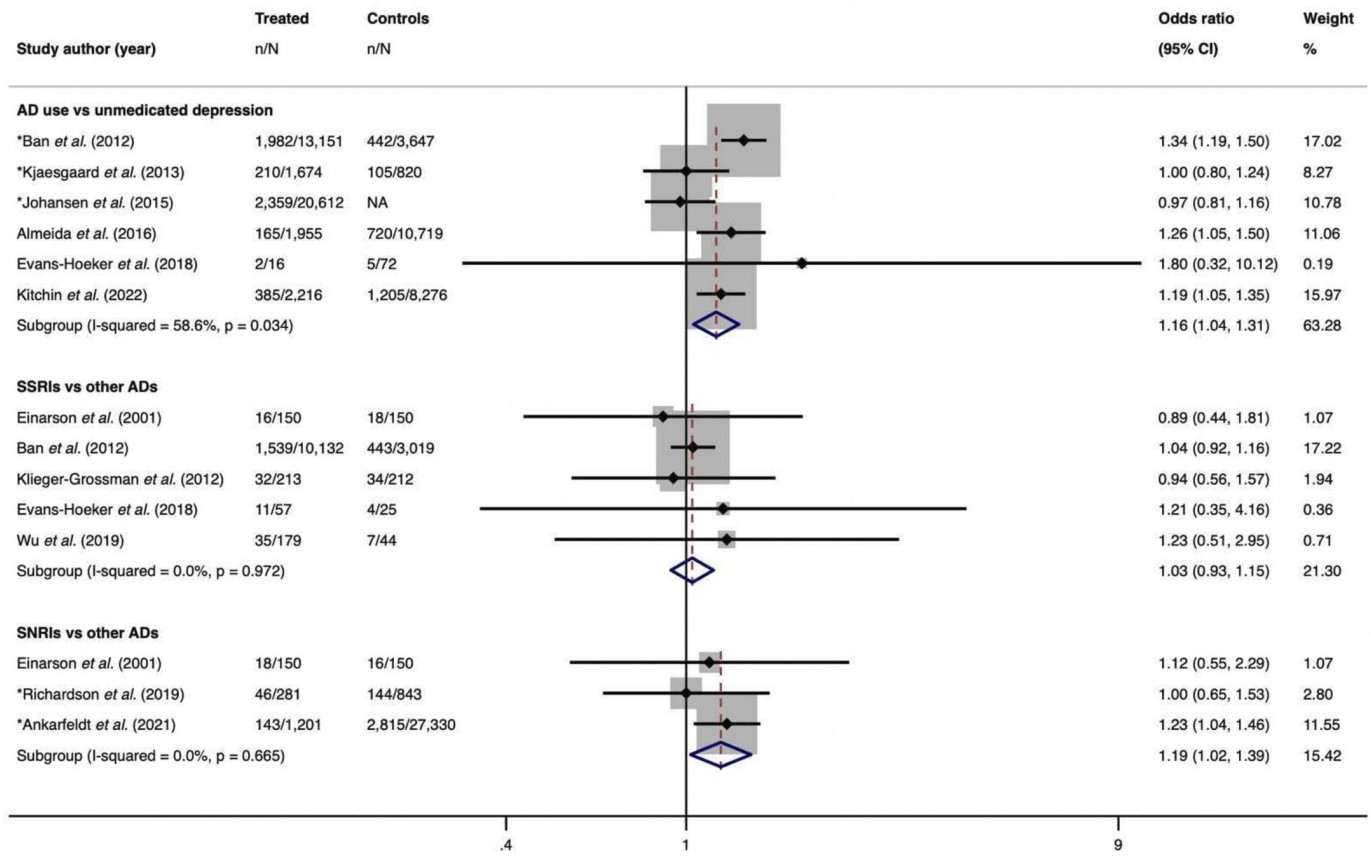

**Figure 3** Three random-effect meta-analysis using depressed cohort comparator. First subgroup assessing any antidepressant use against unmediated depressed individuals. Second subgroup comparing selective serotonin reuptake inhibitors (SSRIs) use and different types of antidepressants (this includes data from papers which compared SSRIs and serotonin-norepinephrine reuptake inhibitors (SNRIs), SSRIs and tricycleric antidepressants (TCAs), SSRIs and any other antidepressants). Third subgroup analysis compared SNRI use and different types of antidepressants (this includes data from papers comparing SNRI and SSRI).

higher summary estimate was observed (pooled OR=1.42 95% CI 1.11 to 1.81, $I^2$=49.6%, p=0.11) (figure 2).

Subgroup analyses investigating SSRIs and SNRIs separately were performed again after restricting to a depressed cohort. When comparing SSRI use and individuals using other antidepressant types, it produced an attenuated summary estimate of 1.03 (95% CI 0.93 to 1.15, $I^2$=0%, p=0.972). Whereas the summary estimate comparing SNRI against other antidepressant use on the outcome of miscarriage was slightly higher (pooled summary estimate=1.19 95% CI 1.02 to 1.39, $I^2$=0%, p=0.665) (figure 3).

## DISCUSSION
### Summary of evidence
This is the largest and most up-to-date review investigating the association between antidepressant use during pregnancy and miscarriage, with around twice as many studies included than previous reviews.[20–22] After pooling best estimates from 29 studies, we found little evidence to suggest an increased risk of miscarriage among antidepressant users after stratifying by maternal depression, with CIs almost spanning the null. The increased risk of miscarriage among pregnant women taking antidepressants compared with unexposed pregnant women in the general population is likely driven by confounding by indication.

### Interpretation of findings
The general population analysis result was comparable to other smaller meta-analyses, which found effect estimates of 1.45, 1.49 and 1.47.[8 20 22] The increased estimates found by these papers are likely driven by fewer included studies, with 11, 14 and 6 respectively, compared with 29 in our study. The elevated effect estimates when using a general population control could likely be driven by confounding by indication, a bias which is lessened when comparing with depressed controls. To the best of our knowledge, meta-analyses in the field investigating miscarriage have not yet used an unexposed depressed comparator; thus, no comparable results exist. As only six studies were used in this subgroup meta-analysis, the summary estimate lacks power and robustness, inducing potential for chance findings, highlighted by large SEs. Additionally, this weak miscarriage risk could

be reflecting depression severity, whereby unexposed depressed individuals have a lower severity of depression than exposed individuals. Studies have indicated miscarriage risk between individuals continuing and discontinuing antidepressant use on start of pregnancy is similar.[18 48] Therefore, future studies should incorporate the comparator of discontinued antidepressant use to allow for more comparable depression severity. Furthermore, other confounding factors that were not adequately controlled for could be biasing this estimate; thus, adjusting for all key confounders is essential in future studies to minimise residual confounding.

Our study found that SNRIs carry a greater risk of miscarriage than SSRIs when compared with an unexposed general population comparator group. No meta-analyses have been conducted on SNRIs to our knowledge, likely due to the higher prevalence of other antidepressant use during pregnancy, for example, SSRIs.[12] Previous reviews using a fixed-effect method found ORs of 1.70 and 1.87 when comparing SSRIs to unexposed general population comparators.[53 54] The lower effect estimate of 1.30 observed in our study may be partially driven by the use of a random-effect model to account for high heterogeneity, as well as the inclusion of an additional eight studies. The elevated point estimate for SNRIs compared with SSRIs may suggest that miscarriage risk differs between antidepressant classes. Alternatively, these estimates could be highlighting confounding by indication, which was further explored in a set of subgroup analyses that stratified by maternal depression. The summary effect estimates between both SSRIs and SNRIs and other antidepressant classes attenuated towards the null, thus, emphasising the potential impact of confounding by indication. The SSRI analyses had consistently lower estimates, including against other antidepressant classes, when comparing pooled estimates with those from the SNRI analyses. The difference in point estimates within the subgroup analysis could be reflecting the differing population types, as SSRIs are used as first line treatment for depression,[55] whereas SNRIs are often used in individuals with more severe depression or when first line options may have not worked.[56] Thus, these results provide evidence for the potential impact of confounding by depression severity, whereby the elevated pooled estimate from the SNRI analysis could be driven by non-exchangeability between individuals who use different classes of antidepressants. Confounding by disease severity was highlighted in the largest included study in the SNRI analysis which discussed that the risk difference observed between duloxetine (SNRI) and SSRI users could be depression severity and residual confounding.[48] However, given the CIs overlap for the subgroup analyses of SSRIs and SNRIs, it is plausible that the true pooled estimates for each analysis are the same. Future network meta-analyses may help elucidate the intricacies of the relationship between individual antidepressant medications or classes and miscarriage.

## Limitations of included studies

The included studies had several limitations. First, only 7% of studies had a moderate overall RoB, due to the limited level of confounding adjustment in papers within the field. Only nine papers adjusted for any variable within all three key confounding domains (online supplemental table S5). Therefore, the included studies may suffer from residual confounding, as strong risk factors for miscarriage, including cotreatments and comorbidities are likely imbalanced between exposure groups,[44] and any effect estimates may be biased away from the null. Additionally, we could not assess by length or severity of depressive disorder as this was rarely stated in the studies.

A second limitation of the included studies is information bias due to exposure misclassification. Exposure was assessed mainly via self-report from physician or individual, or prescription databases. However, only 60% agreement has been found between prescription and self-report data,[18] with conflicting evidence as to the most accurate method to assess exposure to medication.[57 58] Teratogen surveillance data were used in several included studies; however, it poses difficulty as women enquiring about antidepressant exposure are automatically classified as exposed, when they may discontinue use after enquiry. Prescription database error could be less impactful, as it likely suffers from non-differential misclassification and biases the estimate towards the null.[18] This is shown in our sensitivity analysis that was conducted using only prescription database papers whereby results attenuated towards the null (online supplemental table S6), and could potentially be reflecting a more accurate measure of risk. Alternatively, self-report data could be more accurate in terms of ingested drug usage. Nevertheless, sensitivity analyses comparing self-report and prescription database inaccuracies indicate that exposure misclassification does not impact effect estimates significantly, and only reduces precision.[18] Future studies should also look to report number of antidepressant prescriptions as this was rarely collected in previous studies, and is a crucial factor to be investigated.

Thirdly, selection bias is another common methodological limitation in this field. A significant issue observed is that the start of intervention and follow-up do not coincide. This results in conditioning on women who have not experienced a miscarriage between the start of intervention and follow-up but excluding eligible individuals who have. Additionally, exposed individuals are more likely to be in consultation with their healthcare professional when planning a pregnancy to discuss potential risks of antidepressants, compared with unexposed individuals. This results in exposed individuals having a longer observed risk period and as such, more chance of detecting and recording miscarriages, therefore biasing results away from the null.[59] These methodological errors contributing to a bias which occurs when individuals who have already experienced the

outcome of interest at the point of study commencement not being included,[60] referred to as left truncation.[61]

## Strengths and limitations of review

Our review has important strengths not addressed in any previous review investigating miscarriage. Confounding by indication and depression severity was a key part of the analysis and results, elucidating the true independent effect of antidepressant use during pregnancy. As this important bias has not previously been addressed in any other review in the field, it emphasises the requirement for accounting for psychiatric illness in future reviews and studies to avoid biased estimates. To the best of our knowledge, our study was also the first review in this field to assess RoB using the ROBINS-I tool. This tool gave a significant advantage over the commonly used Newcastle-Ottawa scale and highlighted the need for more thorough confounding adjustment and reporting of missing data in future studies. Additionally, the issues surrounding exposure recall bias were minimised using mainly cohort studies, and the included case-control studies having prospective exposure measurement. Finally, 29 studies published up to 2022 were identified and used in the review through an extremely thorough search strategy, thus enhancing statistical power and insight over previous reviews.

The review also had limitations. The studies included could suffer from publication bias as few studies were distributed around the null line and most found associations that transcended the null (online supplemental figure S2). Additionally, small studies producing inverse relationships are absent from the field. This conclusion was supported by the small p value produced in the Egger test. If only positive results are being published, this could result in exaggerated review results. The authors did not include grey literature due to limited resource which could have also contributed to this finding. Alternatively, these conclusions may not be driven by publication bias, but instead either real causal associations or lack of methodological rigour throughout the studies.

High heterogeneity also existed between papers, likely due to the included studies investigating different research questions, implementing different methodologies and populations, timing of exposure and confounding adjustment, thereby reducing the quality of the meta-analysis. Additionally, the lack of available data prevented a trimester specific subgroup analysis. Previous studies have identified first trimester miscarriage risk with antidepressant use to be higher compared with second trimester,[18] indicating antidepressant exposure could be time sensitive. However, this could reflect the time window for highest risk of miscarriage being between weeks 7 and 12, during first trimester.[62] Future reviews should aim to incorporate this subgroup analysis to investigate whether it alters conclusions made.

## CONCLUSION

In conclusion, we did not find any clear evidence to suggest antidepressants increase the risk of miscarriage, though the results should be treated with caution due to the large heterogeneity present. This report highlights the effect of confounding by indication on previous literature. As RCTs cannot be used, future observational studies with larger sample sizes and more robust confounding adjustment, especially in relation to indication for treatment are needed. Triangulating results with differing methodologies should help strengthen causal inference. Additionally, network meta-analyses are needed to fully investigate the differing risk between antidepressant classes. Overall, the present study provides a summary of the most up-to-date findings of miscarriage risk following antidepressant use during pregnancy, to better inform clinicians and women when weighing up treatment for depression during pregnancy.

**Acknowledgements** The views in this research are of the authors and not of the funders. The authors thank the corresponding authors and coauthors of the studies included in this review. We thank Professor Julian Higgins who was extremely helpful with queries involving the ROBINS-I tool.

**Contributors** All authors helped with interpretation of results. SS did the full-text screening, data extraction, risk of bias assessment, statistical analysis and wrote the report. SS was responsible for the overall content as the guarantor. FM did the title and abstract screening, full-text screening, the second review of data extraction and risk of bias assessment, statistical analysis and had input on the report writing. HF did the title and abstract screening, full-text screening, and had input on all other elements of the paper. DR suggested the research question, provided supervision and comment on the manuscript.

**Funding** SS is supported by an NIHR predoctoral fellowship (NIHR301109). FM is supported by the Wellcome Trust (WT 218495/Z/19/Z). HF and DR acknowledge support from the NIH (1R01NS107607). This research was also supported by the NIHR Biomedical Research Centre at the University of Bristol and the University Hospitals Bristol and Weston NHS Foundation Trust.

**Competing interests** None declared.

**Patient and public involvement** Patients and/or the public were not involved in the design, or conduct, or reporting, or dissemination plans of this research.

**Patient consent for publication** Not applicable.

**Ethics approval** Not applicable.

**Provenance and peer review** Not commissioned; externally peer reviewed.

**Data availability statement** Data are available upon reasonable request. Please contact the first author (SS) if you would like to see any data not included in the article or supplementary material.

**ORCID iD**
Sophie Smith http://orcid.org/0000-0002-1248-5477

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
