## [Reviewer comments · BMJ Open]

ARTICLE DETAILS

TITLE (PROVISIONAL)	The association between antidepressant use during pregnancy and miscarriage: A systematic review and meta-analysis
AUTHORS	Smith, Sophie; Martin, Flo; Rai, Dheeraj; Forbes, Harriet

VERSION 1 – REVIEW

REVIEWER	Akhtar, Sohail Government College University Lahore, Statistics
REVIEW RETURNED	29-May-2023

GENERAL COMMENTS	There is a significant publication bias in the meta-analysis based on Funnel plot and Egger test. Publication bias is a serious problem in systematic reviews and meta-analyses, which can affect the validity and generalization of conclusions. It is strongly advised to use trim and fill method and do the recalculation.
--

REVIEWER	Pawliuk, Colleen British Columbia Children's Hospital, Research Institute
REVIEW RETURNED	06-Jun-2023

GENERAL COMMENTS	Please note that I am a health librarian, so my peer review will focus on the methods of the systematic review. Abstract In the Aims section, provide a clear statement of your review question/objective in PECO format. Introduction The Authors mention other reviews on their topic. Where did they search for existing reviews? They should list the sources they searched so the reader is able to understand the comprehensiveness of their search. The Authors should re-phrase their review question/objective using the PECO format so that it is clear. "Among pregnant people, what the association between antidepressant use and miscarriage?" Methods I see the Authors used the MOOSE guidelines to report their review. This should be cited. The Authors could also consider using the PRISMA 2020 Guidelines to report their review as there is more detail in these guidelines. The checklist can be found here: http://www.prisma-statement.org/documents/PRISMA_2020_expanded_checklist.pdf. Did the Authors use any methodological guidance for their review (e.g. Cochrane Handbook)? If so, this should also be cited. The Author should include the platform used for each database. Ideally, they should also include dates of coverage of each database. For example, MEDLINE (Ovid, 1946-2022). This
---

	increases the reproducibility of their methods as date coverage of databases can vary based on subscriptions. The date of last search almost 1.5 years out of date. Have the Authors considered updating the search for their review again? The Authors mention they searched the grey literature. What sources were searched? These should be listed, and if the number is high this list could be included in the supplement. My assumption is that one or more clinical trial registries were searched only. The Author's PROSPERO registration states they will include only peer-reviewed studies. How did they assess this? How did they do this if they search the grey literature, which is generally not peer-reviewed? Did a librarian/information professional perform the searches? This should be included in the description of the search methods. I applaud the Authors for including non-English studies and seeking translation. Did the Authors de-duplicate the results and perform study selection using software, e.g. Covidence? Details of the tools used should be included. Discussion Strengths and limitation of the review "Finally, 26 studies published up to 2021 were identified and used in the review through an extremely thorough search strategy, thus enhancing statistical power and insight over previous reviews." From the reported methods of this systematic review I do not agree that this review had a thorough search strategy. While it's clear that the Authors performed a reasonably good quality searches in the databases, there was little grey literature searching completed (my assumption is only clinical trial registry/registries) and no supplementary search (e.g. citation searching). A more thorough search could have included looking at conference abstracts which we know often are not published as full articles, theses and dissertations and other forms of grey literature. Backward and forward citation searching could locate key studies missed by the search. The Authors also could have undertaken hand searching of key journals or conferences. I believe this lack of comprehensiveness should be included explicitly in the limitations and tied to the appearance of publication bias. A further limitation is the 1.5 year gap since the last search was completed. Supplementary Table S1 Search strategies for each information source should be included, rather than just the MEDLINE search. Ideally the search strategy includes line numbers and is pasted right out of the database when the searches are run. This increases the reproducibility of your systematic review.
--	---

REVIEWER	biffi, annalisa Milan Bicocca University
REVIEW RETURNED	12-Jun-2023

GENERAL COMMENTS	This systematic review and meta-analysis discussed an interesting topic regarding the the association between antidepressant use during pregnancy and miscarriage, however the manuscript has limitations and several points, which need to be clarified. Introduction Could you move the paragraph regarding the "Existing reviews examining...." and the paragraph about "The methodological
---

	limitations..” in the Discussion section? Methods section Could you explain why you have considered the miscarriages as of rare nature? May you list the types of other antidepressants? Results section Is it possible to check about the length of the depressive disorders (or when the disease started)? As well as more information about the antidepressant use and its dose/number of prescriptions? Could you specify the types of cotreatments and comorbidities? May the administration of other medications or the presence of other illness increase the risk the miscarriage? I think it could be useful to report the women’s age and if they had previous miscarriage, other than the presence of further mental disease. May you indicate the weeks’ gestation characterized by the loss of fetus? Could you report the articles’ reference about the considered comparator? Is it possible to specify the type of databases in figure 1 as well as the reports of included studies? Have you verified that studies with the same first author have not included a part or the same population? I have not seen the results regarding the subgroup analyses, which investigate the effect of SSRIs and SNRIs after restricting to a depressed cohort. Could you report the Prisma checklist?
--	--

REVIEWER	Hasen, Aragaw Asfaw Samara University, statistics
REVIEW RETURNED	23-Jun-2023

GENERAL COMMENTS	Comments to the Authors: Thank you for the opportunity to review the manuscript titled “The association between antidepressant use during pregnancy and miscarriage: A systematic review and meta-analysis”. The findings of this systematic review and meta analysis will help to provide a comprehensive evidence on any association between antidepressant use during pregnancy and miscarriage. I rely on there are opportunities to further strengthen this manuscript. Here are the comments to be addressed by the author. 1. Title - The title is interesting, the author should remove the underline from all parts of the manuscript. 2. Abstract -Well written, but needs revision. Write the abstract in the form of objective, design, data sources, eligibility criteria and outcomes, data extraction and synthesis, result and conclusion. -Also in the methods indicate what software is used for the analysis? -Please write the result report in the form of, for instance the pooled (pooled odds ratio = 95%CI: , I2 = , p<0.001). -Results: 26 studies were included in the systematic review and meta-analysis. Antidepressant users initially appeared to have a higher risk of miscarriage compared to unexposed individuals from the general population (pooled odds ratio: 1.29, 95% CI: 1.21-1.38, I2 = 70.8%, p-value<0.001). However, from 5 studies the effect estimate decreased when comparing against unexposed individuals with maternal depression (pooled odds ratio: 1.16, 0.98-1.36; I2 =
---

67.7%, p- value<0.001). 19 studies suffered from serious risk of bias, and two studies had moderate.

-The author should modify the writing format of result in the manuscript as stated above.

-Please add PROSPERO registration number next to conclusion.

-Write the strength and limitation in short and precise sentence?

For instance

“Confounding by indication and depression severity was a key part of the analysis and results, elucidating the true independent effect of antidepressant use during pregnancy. As this important bias has not previously been addressed in any other review in the field, it emphasises the requirement for accounting for psychiatric illness in future reviews and studies to avoid biased estimates.” is so long!

So, the author should re-write as follows:

-Confounding by indication and depression severity was considered in the analysis and results, elucidating the true independent effect of antidepressant use during pregnancy.”

-The risk of bias was assessed by the ROBINS-I tool.

- A large amount of studies were identified through an extremely thorough search strategy, thus enhancing statistical power and insight over previous reviews were strengths.

- High heterogeneity existed between papers, likely due to the included studies investigating different research questions, implementing different methodologies and populations, timing of exposure and confounding adjustment, thereby reducing the quality of the meta-analysis.

- The studies included could suffer from publication bias, and small studies producing inverse relationships are absent from the field were the limitations.

2. Introduction

- Well written.

- Please add the subtitle objective, and the statement in line #41-#42 can be modified by “The main objective of this study is to conduct an updated and thorough systematic review and meta-analysis regarding the association between antidepressant use during pregnancy and miscarriage.”

3. Methods

-Please add the sub heading “Protocol registration” next to Methods. Then the authors should write “This study is conducted in accordance with the Preferred Reporting Items for Systematic Review and Meta Analysis (PRISMA) 2020. The protocol for this meta-analysis was registered in PROSPERO CRD42021262839, available from https://www.crd.york.ac.uk/prospero/display_record.php?ID=CRD42021262839

- Most of the basic components of systematic review and meta analysis are included in the manuscript, it is appreciated. But they should be in the form of the journal (BMJ Open) guideline. So, The authors should add the following respective subtitles accordingly.

Search strategy and information sources page 5, line #51 (as it is)

Eligibility criteria page 6 line #20

- The author’s should clearly state the inclusion and exclusion criteria for this systematic review and meta analysis.

Outcome measures

- The authors clearly state the outcome measures of this study?

Study selection page 6 line # 39

Data extraction, delete data analysis page 6, line #55

Risk of bias assessment page 7, line # 19

Statistical analysis page 7, line # 35

-Move the explanation about protocol to first line of methods section.

	-Patient and Public Involvement -This is the review of studied and published works, Say "No Patient and Public Involvement" -Role of the funding source Delete this, the funding information can describe it. 4. Results - Well written, but needs some modification to increase the clarity of findings. The following subtitles should be included in respective order. Search results Study characteristics Risk of bias assessment - The authors should also clarify how many researcher do this assessment? Publication bias Sensitivity analysis Meta analysis results -Make sure that the consistency in the manuscript writing style as that of stated in the abstract comments? - In the figure 1, remove the statement " reports of included studies(n=)" in the last step. 5. Discussion - Well written. 6. Conclusions - Well written. 7. References - Well written.
--	---

VERSION 1 – AUTHOR RESPONSE

Reviewer: 1

Dr. Sohail Akhtar, Government College University Lahore

Comments to the Author:

There is a significant publication bias in the meta-analysis based on Funnel plot and Egger test. Publication bias is a serious problem in systematic reviews and meta-analyses, which can affect the validity and generalization of conclusions. It is strongly advised to use trim and fill method and do the recalculation.

Thank you for this comment. We have investigated the trim and fill method which looks to remove the smaller studies contributing to the funnel plot asymmetry, then use this reduced funnel plot to estimate the true 'centre' of the funnel, whilst replacing the removed studies around the central funnel. However, this methodology heavily relies on the assumption that there should be a symmetric funnel plot, and no other reasons are considered that may play a role in the asymmetry, other than publication bias. Therefore, this method is known to perform poorly when there is large between-study heterogeneity (1). As highlighted in the manuscript, there is significant between-study heterogeneity in this systematic review, indicated through the high I² values. Therefore, we believe that completing the trim and fill method will not provide any valuable insight here, and instead add uncertainty. We hope the reviewer agrees with our reasoning in this situation.

Reviewer: 2

Ms. Colleen Pawliuk, British Columbia Children's Hospital

Comments to the Author:

Please note that I am a health librarian, so my peer review will focus on the methods of the systematic review.

Abstract

In the Aims section, provide a clear statement of your review question/objective in PECO format. We have included the following sentence in the background section:

“We aimed to conduct a systematic review and meta-analysis of studies among pregnant women regarding the association between exposure to antidepressants during pregnancy and the risk of miscarriage, compared to pregnant women not exposed to antidepressants.”

Introduction

The Authors mention other reviews on their topic. Where did they search for existing reviews? They should list the sources they searched so the reader is able to understand the comprehensiveness of their search.

We examined and looked through existing reviews that appeared in the searches described in the methods section, as well as checked their references for relevant papers in the review.

The Authors should re-phrase their review question/objective using the PECO format so that it is clear. “Among pregnant people, what the association between antidepressant use and miscarriage?” The review question has been re-phrased to include: “We aimed to conduct a systematic review and meta-analysis of studies among pregnant women regarding the association between exposure to antidepressants during pregnancy and the risk of miscarriage, compared to pregnant women not exposed to antidepressants.”

Methods

I see the Authors used the MOOSE guidelines to report their review. This should be cited. The Authors could also consider using the PRISMA 2020 Guidelines to report their review as there is more detail in these guidelines. The checklist can be found here: http://www.prisma-statement.org/documents/PRISMA_2020_expanded_checklist.pdf.

Did the Authors use any methodological guidance for their review (e.g. Cochrane Handbook)? If so, this should also be cited.

We consulted the Cochrane handbook and used the PRISMA 2020 guidelines to report the review. The PRISMA checklist will be uploaded as additional information and the Cochrane handbook is now cited in the manuscript.

The Author should include the platform used for each database. Ideally, they should also include dates of coverage of each database. For example, MEDLINE (Ovid, 1946-2022). This increases the reproducibility of their methods as date coverage of databases can vary based on subscriptions. Ovid was used to search each database. EMBASE (Ovid, 1980 to 2023), MEDLINE (Ovid, 1946 to 2023) and PsycInfo (Ovid, 1806 to 2023). This has been added to the manuscript.

The date of last search almost 1.5 years out of date. Have the Authors considered updating the search for their review again?

The authors have updated their search up to 6th August 2023. This new search found three additional papers, and these new findings have been incorporated into the manuscript.

The Authors mention they searched the grey literature. What sources were searched? These should be listed, and if the number is high this list could be included in the supplement. My assumption is that one or more clinical trial registries were searched only.

Thank you for this comment. On reflection we did not include grey literature and instead screened citations in other papers and reviews. This element of the methods has been removed.

The Author's PROSPERO registration states they will include only peer-reviewed studies. How did they assess this? How did they do this if they search the grey literature, which is generally not peer-reviewed?

As mentioned above, the mention of grey literature was made in error, therefore all of the included studies were peer reviewed.

Did a librarian/information professional perform the searches? This should be included in the description of the search methods.

HF performed the searches and we have noted this in the methods section.

I applaud the Authors for including non-English studies and seeking translation.

Many thanks.

Did the Authors de-duplicate the results and perform study selection using software, e.g. Covidence? Details of the tools used should be included.

We used EndNote to de-duplicate references, this has been added to the methods section.

Discussion

Strengths and limitation of the review

“Finally, 26 studies published up to 2021 were identified and used in the review through an extremely thorough search strategy, thus enhancing statistical power and insight over previous reviews.” From the reported methods of this systematic review I do not agree that this review had a thorough search strategy. While it's clear that the Authors performed a reasonably good quality searches in the

databases, there was little grey literature searching completed (my assumption is only clinical trial registry/registries) and no supplementary search (e.g. citation searching). A more thorough search could have included looking at conference abstracts which we know often are not published as full articles, theses and dissertations and other forms of grey literature. Backward and forward citation searching could locate key studies missed by the search. The Authors also could have undertaken hand searching of key journals or conferences. I believe this lack of comprehensiveness should be included explicitly in the limitations and tied to the appearance of publication bias.

We believe that the search strategy was thorough, we did complete citation searching, specifically backward citation searching, as well as including all conference abstracts relating to the topic. We believe the dominating reason why publication bias is present is due to the nature of pregnancy research. Albeit we do agree with the reviewer and acknowledge that negative or null papers were only available in grey literature and not published. Therefore, we have added a comment in the limitations section of the manuscript in that further citation searching could have been conducted, but due to limited resource we couldn't. However, we do not anticipate the results changing dramatically through inclusion of grey literature.

A further limitation is the 1.5 year gap since the last search was completed.

This has been addressed by updating the search.

Supplementary Table S1

Search strategies for each information source should be included, rather than just the MEDLINE search. Ideally the search strategy includes line numbers and is pasted right out of the database when the searches are run. This increases the reproducibility of your systematic review.

This information has been added and included in Supplementary Table S1.

Reviewer: 3

Dr. annalisa biffi, Milan Bicocca University

Comments to the Author:

This systematic review and meta-analysis discussed an interesting topic regarding the the association between antidepressant use during pregnancy and miscarriage, however the manuscript has limitations and several points, which need to be clarified.

Introduction

Could you move the paragraph regarding the "Existing reviews examining...." and the paragraph about "The methodological limitations.." in the Discussion section?

Amended in line with this comment.

Methods section

Could you explain why you have considered the miscarriages as of rare nature?

The reason we said it was rare was because we pooled ratio measures together and believed that it was 'rare enough' to use this methodology. We have since conducted a sensitivity analysis in the general population section of the analysis whereby we used odds ratios only as most of the ratios given were these and removed the risk and hazard ratios. This produced similar results to the primary analyses and can be found in the supplementary figure (S2 Figure). We didn't conduct further sensitivity analyses utilising hazards and risk due to the small numbers of studies resulting in the analysis being under-powered. However, we believe this sensitivity analysis indicates our methodology of pooling ratios doesn't affect results. Aside from the methodology, in general we do not believe that miscarriages are rare and this has been removed from the manuscript.

May you list the types of other antidepressants?

Details of the other antidepressants utilised as the comparator in the depressed stratified meta-analysis can be found in the footnote of figure 3.

Results section

Is it possible to check about the length of the depressive disorders (or when the disease started)? As well as more information about the antidepressant use and its dose/number of prescriptions?

Some of these papers report the binary measure of previous depressive disorder, however, unfortunately none of the papers reported length of depressive disorder or when the disease started, therefore, this will not be able to be included. Additionally, only some of the papers used in the review had an identification process through prescription tracking, a lot of the papers instead used self-report or tracking through enquiry to an information helpline. Only one paper mentioned collecting information around the number of prescriptions (Ban et al, 2012). They categorised individuals

depending on whether they have received a repeat prescription for a class of drug in the first trimester. However, they didn't report the results for these two subgroups. Therefore, unfortunately again this investigation cannot be completed. These limitations have been added to the manuscript. Could you specify the types of cotreatments and comorbidities? May the administration of other medications or the presence of other illness increase the risk the miscarriage?

Many cotreatments and comorbidities would have existed within the population of individuals in these studies. Both factors could have influenced and increased the risk of miscarriage. This has been added to the manuscript.

I think it could be useful to report the women's age and if they had previous miscarriage, other than the presence of further mental disease.

The mean age of women in each study will be added to Table S3 and provided. Only two papers (Andersen, 2013 & Johansen, 2015) collected data on previous miscarriage. This was used in the adjustments for statistical analysis.

May you indicate the weeks' gestation characterized by the loss of fetus?

This was very rarely stated in papers, however, for the papers where it was stated, this was mentioned in the results section here: "The miscarriage definition was rarely stated, however, ranged from loss of fetus at either 20 (n=3) (Almeida et al., 2016; Nakhai-Pour et al., 2010; Wu et al., 2019), 22 (n=2) (Johansen et al., 2015; Kjaersgaard et al., 2013), or 24 weeks' (n=1) (Richardson et al., 2019) gestation. Now included in manuscript.

Could you report the articles' reference about the considered comparator?

The comparators utilised by each article were list in the supplementary materials (s4 table)

Is it possible to specify the type of databases in figure 1 as well as the reports of included studies?

The databases mentioned in Figure 1 are described in the methods section whereby Medline, PsychInfo and Embase were searched.

Have you verified that studies with the same first author have not included a part or the same population?

Yes, this was checked during the data extraction stage.

I have not seen the results regarding the subgroup analyses, which investigate the effect of SSRIs and SNRIs after restricting to a depressed cohort.

This was reported in the supplementary material (S4 table)

Could you report the Prisma checklist?

Yes. It will be included as an additional supplementary material.

Reviewer: 4

Mr. Aragaw Asfaw Hasen, Samara University

Comments to the Author:

Comments to the Authors:

Thank you for the opportunity to review the manuscript titled "The association between antidepressant use during pregnancy and miscarriage: A systematic review and meta-analysis". The findings of this systematic review and meta analysis will help to provide a comprehensive evidence on any association between antidepressant use during pregnancy and miscarriage. I rely on there are opportunities to further strengthen this manuscript. Here are the comments to be addressed by the author.

1. Title

- The title is interesting, the author should remove the underline from all parts of the manuscript. Done.

2. Abstract

-Well written, but needs revision. Write the abstract in the form of objective, design, data sources, eligibility criteria and outcomes, data extraction and synthesis, result and conclusion.

Thanks for raising this, we have revised the abstract to fit with the requirements of the BMJ Open.

-Also in the methods indicate what software is used for the analysis?

Analyses were conducted using STATA, this is listed in the methods section.

-Please write the result report in the form of, for instance the pooled (pooled odds ratio = 95%CI: I² = , p<0.001).

-Results:

26 studies were included in the systematic review and meta-analysis. Antidepressant users initially appeared to have a higher risk of miscarriage compared to unexposed individuals from the general population (pooled odds ratio: 1.29, 95% CI: 1.21-1.38, I² = 70.8%, p-value<0.001). However, from 5

studies the effect estimate decreased when comparing against unexposed individuals with maternal depression (pooled odds ratio: 1.16, 0.98-1.36; I² = 67.7%, p-value < 0.001). 19 studies suffered from serious risk of bias, and two studies had moderate.

-The author should modify the writing format of result in the manuscript as stated above.
Rectified in manuscript in line with above request.

-Please add PROSPERO registration number next to conclusion.

The PROSPERO registration number has been added next to the conclusion. Please find below:
PROSPERO 2021 CRD42021262839 Available from:

https://www.crd.york.ac.uk/prospero/display_record.php?ID=CRD42021262839

-Write the strength and limitation in short and precise sentence?

For instance

“Confounding by indication and depression severity was a key part of the analysis and results, elucidating the true independent effect of antidepressant use during pregnancy. As this important bias has not previously been addressed in any other review in the field, it emphasises the requirement for accounting for psychiatric illness in future reviews and studies to avoid biased estimates.” is so long! So, the author should re-write as follows:

-Confounding by indication and depression severity was considered in the analysis and results, elucidating the true independent effect of antidepressant use during pregnancy.”

-The risk of bias was assessed by the ROBINS-I tool.

- A large amount of studies were identified through an extremely thorough search strategy, thus enhancing statistical power and insight over previous reviews were strengths.

- High heterogeneity existed between papers, likely due to the included studies investigating different research questions, implementing different methodologies and populations, timing of exposure and confounding adjustment, thereby reducing the quality of the meta-analysis.

- The studies included could suffer from publication bias, and small studies producing inverse relationships are absent from the field were the limitations.

Thank you, adjusted accordingly.

2. Introduction

- Well written.

- Please add the subtitle objective, and the statement in line #41-#42 can be modified by “The main objective of this study is to conduct an updated and thorough systematic review and meta-analysis regarding the association between antidepressant use during pregnancy and miscarriage.”
Amended.

3. Methods

-Please add the sub heading “Protocol registration” next to Methods. Then the authors should write “This study is conducted in accordance with the Preferred Reporting Items for Systematic Review and Meta Analysis (PRISMA) 2020. The protocol for this meta-analysis was registered in PROSPERO CRD42021262839, available from

https://www.crd.york.ac.uk/prospero/display_record.php?ID=CRD42021262839

Completed.

- Most of the basic components of systematic review and meta analysis are included in the manuscript, it is appreciated. But they should be in the form of the journal (BMJ Open) guideline. So, The authors should add the following respective subtitles accordingly.

Search strategy and information sources page 5, line #51 (as it is)

Eligibility criteria page 6 line #20

- The author’s should clearly state the inclusion and exclusion criteria for this systematic review and meta analysis.

Outcome measures

- The authors clearly state the outcome measures of this study?

Study selection page 6 line # 39

Data extraction, delete data analysis page 6, line #55

Risk of bias assessment page 7, line # 19

Statistical analysis page 7, line # 35

-Move the explanation about protocol to first line of methods section.

-Patient and Public Involvement

-This is the review of studied and published works, Say "No Patient and Public Involvement"

-Role of the funding source

Delete this, the funding information can describe it.

Great, thank you for this comment. Structure has been amended to align with this comment. As the description on outcome measures is so small, this is listed under eligibility criteria.

4. Results

- Well written, but needs some modification to increase the clarity of findings. The following subtitles should be included in respective order.

Search results

Study characteristics

Risk of bias assessment

- The authors should also clarify how many researcher do this assessment?

Publication bias

Sensitivity analysis

Meta analysis results

-Make sure that the consistency in the manuscript writing style as that of stated in the abstract comments?

- In the figure 1, remove the statement " reports of included studies(n=)" in the last step.

Thank you, order of results changed in line with the above comments. To confirm, two researchers independently completed the risk of bias assessment, this is mentioned in methods section.

5. Discussion

- Well written.

6. Conclusions

- Well written.

7. References

- Well written.

Thank you.

References.

1. [https://handbook-5-](https://handbook-5-1.cochrane.org/chapter_10/10_4_4_2_trim_and_fill.htm#:~:text=The%20basis%20of%20the%20method.around%20the%20centre%20(filling).)

[1.cochrane.org/chapter_10/10_4_4_2_trim_and_fill.htm#:~:text=The%20basis%20of%20the%20method.around%20the%20centre%20\(filling\).](https://handbook-5-1.cochrane.org/chapter_10/10_4_4_2_trim_and_fill.htm#:~:text=The%20basis%20of%20the%20method.around%20the%20centre%20(filling).)

VERSION 2 – REVIEW

REVIEWER	biffi, annalisa Milan Bicocca University
REVIEW RETURNED	13-Oct-2023

GENERAL COMMENTS	Thank you for the changes made on your manuscript. I have some few questions about: Results section -Is it possible to perform a stratified analysis based on the methods of prescription identification (questionnaire, database..)? -May you perform a subgroup analysis depending on the presence of adjustments (yes vs no)? -Could you report the related reference of the mentioned articles in this sentence: "All studies had a comparator group of unexposed individuals without depression; some papers used additional comparator groups including depressed pregnant women exposed to an alternative antidepressant to the primary specified type
---

	(REFERENCE?), unexposed depressed pregnant women,(17, 18, 37, 42, 43, 50) and pregnant individuals using antidepressants whereby the indication for treatment wasn't depression.(17, 32, 43)" -In the last revision, I have asked for the subgroup analysis which investigates the effect of SSRIs and SNRIs after restricting to a depressed cohort, and you indicated me the Supplementary Table S4. I think I was not clear. Could you perform a stratified analysis to investigate the effect of SSRIs and SNRIs in the depressed cohort? Discussion section -Could you add in the Limitation section, the severity of depressive disorders (as you correctly specified the length of depressive disorders)?
--	--

VERSION 2 – AUTHOR RESPONSE

We would like to thank the editor for the opportunity to submit a minor revision on our manuscript and to Dr Biffi for their time carefully considering our study and providing useful comments for improvement. Please see our response for each comment below.

Dr. annalisa biffi, Milan Bicocca University

Comments to the Author:

Thank you for the changes made on your manuscript. I have some few questions about:

Results section

-Is it possible to perform a stratified analysis based on the methods of prescription identification (questionnaire, database..)?

Thank you for this suggestion. We have performed a stratified meta-analysis including studies that utilised prescription databases to define the exposure, which is shown in supplementary table S6. The results attenuate towards the null compared to the primary meta-analysis, including all exposure classification methods.

We decided against running a separate meta-analysis for TIS/questionnaire studies for two reasons. The first is that TIS studies are variable in relation to exposure acquisition: some reports are made by clinicians only (e.g., UKTIS), and others come from the public as well as clinicians (e.g., The Motherisk Program). The second is that we can infer what the findings would have been by comparing the findings from the prescription-only subgroup meta-analysis and the primary meta-analysis. We have summarised the findings of this subgroup meta-analysis in the sensitivity analysis of the results section "Additionally, we performed a meta-analysis utilising papers whereby exposure data was collected only through prescription databases. These results highlighted an attenuation towards the null compared to the primary meta-analysis utilising all exposure classification methods (supplementary table S6). Additionally, we discussed the implications of the findings in the limitations of the included studies section of the discussion, "This is shown in our sensitivity analysis that was conducted utilising only prescription database papers whereby results attenuated towards the null (Supplementary table S6), and could potentially be reflecting a more accurate measure of risk."

-May you perform a subgroup analysis depending on the presence of adjustments (yes vs no)?

Thank you for your question. This subgroup analysis was performed in the previous version, through a meta-analysis analogous to this question. Instead of using "presence of adjustment yes/no", we have leveraged the findings from the risk of bias assessment to guide this analysis. Those studies with no confounding adjustment were deemed "Critical" in the risk of bias assessment and were omitted from the meta-analysis and results of this can be found in Supplementary Table S6. We have added some more explanation on this in the sensitivity analysis part of the results section, "Papers with an overall critical RoB or inadequate information to determine their bias category were removed (n=5). The papers removed were deemed critical as they had no confounding adjustments. Results

remained very similar, with a change in the overall estimate of 0 and 0.01 in all AD and SSRIs respectively in the general population after removing the critical papers. Additionally, in the SNRI general population there was a change in the overall estimate of 0.25 after removing the papers with no adjustments (Supplementary Table S6).”

-Could you report the related reference of the mentioned articles in this sentence: “All studies had a comparator group of unexposed individuals without depression; some papers used additional comparator groups including depressed pregnant women exposed to an alternative antidepressant to the primary specified type (REFERENCE?), unexposed depressed pregnant women,(17, 18, 37, 42, 43, 50) and pregnant individuals using antidepressants whereby the indication for treatment wasn’t depression.(17, 32, 43)”

Thank you for highlighting this, these references have now been added.

-In the last revision, I have asked for the subgroup analysis which investigates the effect of SSRIs and SNRIs after restricting to a depressed cohort, and you indicated me the Supplementary Table S4. I think I was not clear. Could you perform a stratified analysis to investigate the effect of SSRIs and SNRIs in the depressed cohort?

Apologies, after I submitted the revisions response, I needed to change the labelling of the materials due to a change of order in the main document. The results of the stratified analysis investigating SSRIs and SNRIs in the depressed cohort can be found in the results table and corresponding figure (supplementary table S6 and figure 3). This analysis took only papers that reported results of SSRI or SNRI against other classes of antidepressant (this is defined within the figure 3). There was only one paper which reported data for a specific antidepressant type against unmedicated depressed individuals (Ban et al., 2012). This paper found a risk ratio of 1.4 (95% CI:1.2-1.7) when comparing SSRIs against unmedicated depressed individuals. However, due to only one study reporting this, it is not possible to meta-analyse. I hope that the current meta-analyses investigating SSRIs and SNRIs against other antidepressant types satisfies your request.

Discussion section

-Could you add in the Limitation section, the severity of depressive disorders (as you correctly specified the length of depressive disorders)?

Thank you for this point; it has been added, “Additionally, we couldn’t assess by length or severity of depressive disorder as this was rarely stated in the studies.”

VERSION 3 – REVIEW

REVIEWER	biffi, annalisa Milan Bicocca University
REVIEW RETURNED	20-Nov-2023
GENERAL COMMENTS	None, thank you for your changes